# Daptomycin for Treatment of *S. Epidermidis* Endocarditis in an Extremely Preterm Neonate—Outcome and Perspectives

**DOI:** 10.3390/children9040457

**Published:** 2022-03-24

**Authors:** Chiara Minotti, Ilaria Zuccon, Elena Priante, Luca Bonadies, Costanza Di Chiara, Daniele Donà, Eugenio Baraldi, Paola Costenaro

**Affiliations:** 1Department of Women’s and Children’s Health, University of Padua, Via Giustiniani 2, 35128 Padua, Italy; ilaria.zuccon@gmail.com (I.Z.); costanza.dichiara@phd.unipd.it (C.D.C.); 2Neonatal Intensive Care Unit, Department of Women’s and Children’s Health, University Hospital of Padua, Via Giustiniani 2, 35128 Padua, Italy; elena.priante@aopd.veneto.it (E.P.); luca.bonadies@aopd.veneto.it (L.B.); eugenio.baraldi@unipd.it (E.B.); 3Division of Pediatric Infectious Diseases, Department of Women’s and Children’s Health, University of Padua, Via Giustiniani 2, 35128 Padua, Italy; daniele.dona@unipd.it (D.D.); paola.costenaro@phd.unipd.it (P.C.)

**Keywords:** endocarditis, *Staphylococcus epidermidis*, daptomycin, neonate, preterm

## Abstract

With a considerable morbidity and mortality burden, infective endocarditis still represents a challenge for clinicians. This is a case of persistent *Staphylococcus epidermidis* endocarditis in an extremely preterm newborn. The infection, initially treated with vancomycin, was successfully cured with daptomycin. Its use was safe and effective, ensuring a complete remission without adverse effects.

## 1. Introduction

Infective endocarditis (IE) may affect subjects with congenital or acquired valvulopathy or intracardiac devices. However, IE is also described in patients without predispositions [1,2]. Its incidence is estimated to be 3.3 per 100,000 annually for infants younger than one year, with 0.05–0.12 estimated cases per 1000 pediatric hospital admissions [3]. Infections by staphylococcal species are becoming more widespread compared to Streptococcal [4]. Several predisposing factors are recognized, including congenital cardiac diseases, prolonged hospitalization in Intensive Care Units (ICUs), central venous lines, prematurity, and prolonged parenteral nutrition [5]. Children and neonates may be affected, and due to a greater difficulty to isolate pathogens from blood cultures and considering that Duke’s criteria weren’t validated for this population, diagnosis and treatment can be challenging [5]. Treatment options are limited in cases of vancomycin failure. Even though off-label, due to limited experience and insufficient literature, daptomycin should be considered for prolonged therapy for its bactericidal effect and good biofilm penetration profile. We aim to report our experience with daptomycin to treat *S. Epidermidis* IE in an extremely preterm newborn.

## 2. Case Report

A male newborn of 24 weeks and five days of gestation (with a birth weight of 550 g) was admitted to the Neonatal ICU of Padua University Hospital after an emergent C-section due to fetal bradycardia. He required resuscitation, positive pressure ventilation, and endotracheal intubation (his Apgar score was 1-1-4-5-6). Due to respiratory distress syndrome, he required surfactant administration and high-frequency ventilation for 15 h. He was then shifted to conventional ventilation and, on day of life (DOL) 3, to non-invasive ventilation (nIPPV) until DOL 16. His ecocardiography on DOL1 was normal. Ampicillin and netilmicin were administered for six days since birth. His workup was negative for early-onset sepsis. He received donated breast milk since birth via an orogastric tube and parenteral nutrition for a month. An umbilical venous catheter was positioned on the first DOL and held in place for six days, and then an epicutaneo-caval catheter was placed.

On DOL 18, while the patient was in nasal continuous positive airway pressure, a sudden clinical worsening occurred, which included marbled skin, frequent apneas, dyspnea, increased oxygen requirement, and a rise of C-reactive protein (CRP). He was initially shifted to a non-invasive neutrally adjusted ventilator assist and eventually reintubated. In suspicion of late-onset sepsis (LOS), blood and mini bronchoalveolar aspirate cultures were ordered, the central venous line was removed after being in place for 16 days, and a lumbar puncture was performed before beginning the empirical administration of intravenous (IV) vancomycin (at a 15 mg/kg/dose every 12 h) and ceftazidime (at a 30 mg/kg/dose every 12 h). The first blood and catheter tip cultures showed positive results for both Methicillin-resistant *Staphylococcus epidermidis* (MRSE) and *S. capitis* susceptible to vancomycin (in which the minimum inhibitory concentration was as such: MIC ≤ 0.5). Despite receiving a targeted treatment (which included vancomycin trough levels of 10 to 25 mg/L) after a transitory clinical amelioration, a further raise of his CRP levels with a septic appearance was observed five days later and included an extremely low platelet count, resulting in required transfusions. Therefore, at DOL 26, a broader antibiotic treatment (which included meropenem instead of ceftazidime) and an antifungal coverage of micafungin were started soon after repeating the blood cultures. Multiple cultures repeated on DOL 25, 27, and 31 showed positive results for MRSE with increased vancomycin MIC (in which the MIC = 2) despite adequate vancomycin trough levels. Figure 1 shows the timeline of the antibiotic course and positive cultures, while the main clinical and laboratory findings are summarized in Table 1.

Possible differential diagnoses, considering the persistent positivity of the blood cultures and elevated CRP levels, were sepsis by a different pathogen or alternative foci of the infection (abscess, meningitis, pneumonia, or endocarditis). An echocardiography was performed on DOL 33, which detected hyperechoic lesions on the mitral valve tensor apparatus on the left ventricle endocardium in the septal–apical area and on the aortic valve, still with a normal function.A round, hyperechoic lesion of a 2.5 mm-diameter was particularly evident on the commissure between the right coronary and non-coronary cusps (Images available in Appendix A). Such lesions were all well identifiable in several projections, confirming the diagnosis of infective endocarditis (two of Duke’s major criteria, Table 1).

An infectious diseases consultation was performed, and vancomycin was promptly discontinued for both the clinical and microbiological failure related to an observed vancomycin-intermediate susceptibility with an antibiotic failure to penetrate the endocardial and valvular biofilm. On DOL 34, the rapid bactericidal drug, daptomycin, was started off-label (in which the MIC = 0.5 and the dosage was 6 mg/kg twice a day) for its safety and good penetration profile in biofilm and following the available data from the previous studies in the literature [6]. Linezolid was not considered because of its bacteriostatic effect and the risk of myelosuppression from its prolonged course in an extremely preterm, immunosuppressed infant [7]. After clinical improvement, the neonate was extubated on DOL 36 and shifted to nIPPV with a maximum of 40% FiO_2_, which was progressively reduced to 21% on DOL 37. On DOL 38, meropenem was discontinued. The respiratory support was discontinued at DOL 65.

The first negative blood culture was found on DOL 37 at the same time CRP became persistently negative. At weekly follow-ups to monitor daptomycin side-effects, creatine phosphokinase (CPK) and renal function were always normal.

The presence of cerebral septic emboli was excluded at subsequent cranial ultrasonographies, while splenic, hepatic and renal embolic lesions were excluded by an abdominal ultrasound. The chest X-rays were always negative for lung consolidations. Subsequent echocardiographies were performed after the daptomycin implementation, which still detected the previous lesion on the aortic valve. For this reason, the daptomycin treatment was prolonged, consistently showing an optimal tolerance profile for six weeks until a mild hypereosinophilia was documented. Since eosinophilic pneumonia is described as a rare complication during treatment with daptomycin, the drug was discontinued. The patient showed no clinical worsening nor lung lesions and was, by then, breathing spontaneously, without the need for oxygen, and thriving on full enteral feeding. After the discontinuation of daptomycin, echocardiography was repeated, which showed a small, hyperechoic lesion on the aortic valve, compatible with fibrosis. Four weeks after the discontinuation of therapy, there were no identifiable residual valvular lesions. CRP, the complete blood count (CBC), and the biochemical profile were normal at the weekly follow-ups. The newborn was then transferred to a peripheral hospital to achieve autonomy in bottle/breastfeeding at 37 weeks of life without bronchopulmonary dysplasia [8] nor other prematurity-related diseases. At the 5-month neonatological follow-up, the infant showed a complete recovery and was in good clinical conditions, which included normal psychomotor development and proportionate growth.

## 3. Discussion

We reported the safe and effective use of daptomycin, started for the clinical and microbiological failure of vancomycin, in an extremely preterm newborn with CoNS LOS and IE. In cases of persistent bacteremia and elevated CRP levels in a central line-associated bloodstream infection (CLABSI) with associated LOS/septic shock, IE should be considered, even in extremely preterm newborns, considering the clinical and laboratory presentation in a fragile, preterm patient after receiving antibiotic therapy at birth for early-onset sepsis (EOS) prevention, total parenteral nutrition, and having a long-term central venous catheter. Daptomycin is a lipopeptide used at high doses (10 mg/kg/day) in cases of *S. aureus* bacteraemia and right-sided IE in adult populations; however, the lack of evidence still limits its use in invasive endovascular neonatal infections [9].

According to the 2015 European Society of Cardiology (ESC) Guidelines for the management of IEs, the recommended antibiotic treatment for methicillin-resistant staphylococci in pediatric patients is IV vancomycin (with 40 mg/kg/day in 2-3 doses) [5]. Vancomycin-intermediate *S. aureus* (VISA) strains often appear in patients with methicillin-resistant *S. aureus* (MRSA) infection after being exposed to a prolonged vancomycin treatment and resulting in antibiotic selective pressure [10]. Recently, decreased vancomycin susceptibility with increasing MICs has also been reported for CoNS; again, probably after selecting resistant strains because of the pressure of antibiotic exposure leading to treatment failures [10]. Studies on *S. aureus* and CoNS IE in adults [9,10,11,12,13] reported that daptomycin and vancomycin are both similarly effective, showing even better outcomes for daptomycin, including survival according to two papers on MRSA bacteremia with elevated vancomycin MICs (0.1 mg/L). Daptomycin is bactericidal and active against bacterial cell membranes with concentration-dependent kinetics. It has good viability within the biofilm, showing a good penetration profile and overcoming one of the major limitations of other antibiotic IE treatments. Its safety profile is generally favorable; periodic CBC and biochemical profile evaluations are required to monitor its rare side-effects (which include hepatotoxicity and renal damage due to rhabdomyolysis and, infrequently, eosinophilic pneumonia) [9]. The recommended pediatric dose is 10 mg/kg/day IV, but the appropriate neonatal dosage is still unclear. Moreover, serum trough levels are not regularly monitored in neonatal settings, nor were they in our case.

There are, however, several cases in the literature that report the uncomplicated use of daptomycin, even in very preterm infants [6,14,15,16,17] (Table 2) with cases of invasive infections or bacteremias by MRSA or MRSE. In these cases, it shows safety and efficacy at superior doses (6 mg/kg/dose) with shorter intervals (twice daily) than those suggested for older children in the 2015 ESC guidelines. Mohzari et al. recently reported a cohort of 21 neonates, mostly preterm (95%) and with very low birth weight (Table 2), receiving a second-line treatment with daptomycin (with 95% as monotherapy) at variable doses (10 mg/kg/day or 6 mg/kg twice daily) after vancomycin’s failure to treat CoNS invasive infections, especially due to *S. Epidermidis* (15/21 cases). In total, thirteen patients recovered and eight died. Nine (42.9%) were cases of IE treated for a median of twenty-two days; five neonates were cured and four died. The cure rate was lower compared to the previous reports, and IE especially showed the highest failure rate, probably after the development of rapid, irreversible congestive heart failure before starting daptomycin [17]. Daptomycin was, otherwise, effective and well-tolerated at 6 mg/kg twice a day and was not to be delayed, ensuring better cure rates. Its safety profile has not been determined yet, but we were able to treat the infection with a clinical and microbiological cure before being forced to stop therapy due to a side effect of the medication. 

## 4. Conclusions

In cases of persistent bacteremia and elevated CRP levels, IE is a possible diagnosis, even in extremely preterm newborns. Indeed, blood cultures should be performed in order to allow a prompt initiation of a microbiologically guided antibiotic therapy, which improves the efficacy and avoids selective pressure and the emergence of antibacterial resistance. Although vancomycin still represents the treatment of choice for methicillin-resistant Staphilococcal neonatal infections, in order to avoid treatment failure in high vancomycin MICs settings, daptomycin is an effective option, even in cases of extremely preterm neonates. Further studies are necessary, including pharmacokinetic evaluations to establish well-defined neonatal doses and randomized-controlled trials to confirm the efficacy and safety of daptomycin in the most vulnerable populations, such as infants and neonates, and including prematures.

## Figures and Tables

**Figure 1 children-09-00457-f001:**
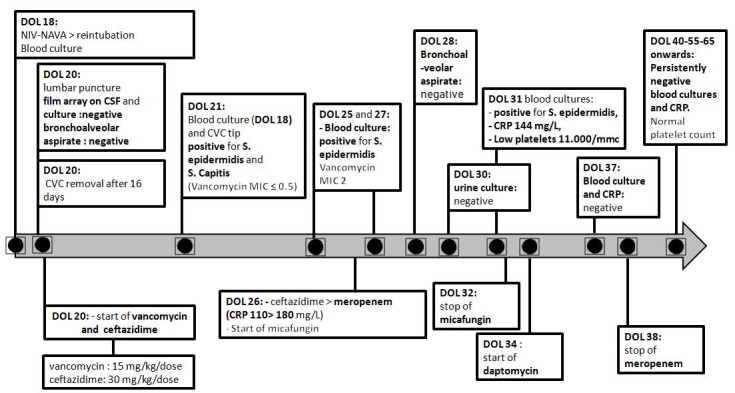
Timeline of antibiotic course and positive cultures. DOL: dav of life, NIV-NAVA: non-invasive neutrally adjusted ventilator assist, CSF: cerebrospinal fluid, CVC: central venous catheter, MIC: minimum inhibitory concentration, CRP: C-reactive protein.

**Table 1 children-09-00457-t001:** Main clinical/laboratory findings and fulfilled criteria for IE.

Clinical Presentation	Laboratory Findings	Criteria for IE
marbled skin	CRP elevation (maximum 180 mg/L) Low PLTs count (11.000/mmc), requiring transfusion	Positive blood cultures for IE (major, Duke’s criteria)
apnoeas		
dyspnoea		
increased oxygen requirement		Echocardiogram identifying IE (major, Duke’s criteria)
invasive ventilation requirement		
Legend: CRP, C-reactive protein; PLTs, platelets; IE, infective endocarditis.		

**Table 2 children-09-00457-t002:** Studies reporting safe and effective daptomycin use in preterm newborns.

Author, Year.	Patient Number, GA (W D/7)	BW (g)	Isolated Bacteria	Clinical Features	Daptomycin Dose	Therapy Duration	Adverse Events	Outcome
Sarafidis et al. [5] 2010	1, 27 3/7	1150	*S. epidermidis*, *E. faecium*	Bloodstream infection/bacteremia	6 mg/kg every 12 h	17 days	No	Recovered and discharged
Hussain et al. [13] 2011	1, 27 4/7	980	MRSA	Bacteremia	10 mg/kg once daily, then 15 mg/kg once daily	14 days	No	Recovered and discharged (No relapse at 6-month follow-up)
Gawronski et al. [14] 2015	1, 24 1/7	480	*S. epidermidis* (MRSE)	Bacteremia	6 mg/kg every 12 h	3 weeks	No	Recovered
Chan et al. [15] 2020	1, 28 1/7	1050	MRSA	Bacteremia, atrio-caval thrombosis	6 mg/kg every 12 h	6 weeks	No	Recovered and discharged (No relapse at 6-month follow-up)
Mohzari et al. [16] 2021	21, 27 (median)	870 (median)	*S. epidermidis*, *S. saprophyticus*, *S. haemolyticus*, *S. capitis*, *S. homins*	IE (*n* = 9) LOS (*n* = 7) EOS (*n* = 3) Bacteremia (*n* = 1) Endovasculitis (*n* = 1)	6 mg/kg every 12 h 10 mg/kg once daily	22 days (median)	No	13 recovered 8 died (GPI: 4 GNI: 3 Abd perf: 1)
Minotti et al. 2021	1, 24 5/7	550	*S. epidermidis* (MRSE)	LOS and IE	6 mg/kg every 12 h	6 weeks	No	Recovered and discharged (No relapse at 5-month follow-up)

Abbreviations: GA: gestational age, BW: birth weight, MRSA: Methicillin-resistant *S. aureus*, MRSE: Methicillin-resistant *S. Epidermidis*, LOS: late-onset sepsis, IE: infective endocarditis; GPI: Gram positive infection; GNI: Gram negative infection; Abd perf: abdominal perforation.

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
