# Peer review of "Daptomycin for Treatment of S. Epidermidis Endocarditis in an Extremely Preterm Neonate—Outcome and Perspectives"

_children, 2022, doi:10.3390/children9040457_

Round 1

Reviewer 1 Report

Line 86 refers to the coronary cusps as coronaric flaps. I would suggest using the normal and accepted nomenclature for this. 

Line 143, what is heosynophylic pneumonia?

You say It is safe to use daptomycin, yet your patient got one of the rare complications with eosinophilic pneumonia. Sure this happened at a time when you could stop the therapy. I'm not saying its not safe, but maybe something can be stated that the safety profile has not been determined yet, but you were able to treat the infection sufficiently before being forced to stop therapy due to a side effect of the medication. 

Otherwise i think this was well written and is useful for this population. 

The Echo pictures are a little disappointing. They may be the best you have, but better echo pictures would be better. 

Author Response

Reviewer 1

Line 86 refers to the coronary cusps as coronaric flaps. I would suggest using the normal and accepted nomenclature for this. 

R1Q1: We modified accordingly in the text, as suggested.

Line 143, what is heosynophylic pneumonia?

R1Q2: We misspelled “eosinophilic” and we corrected accordingly in the manuscript.

You say It is safe to use daptomycin, yet your patient got one of the rare complications with eosinophilic pneumonia. Sure this happened at a time when you could stop the therapy. I'm not saying its not safe, but maybe something can be stated that the safety profile has not been determined yet, but you were able to treat the infection sufficiently before being forced to stop therapy due to a side effect of the medication. 

R1Q3: As reported in the manuscript, in our patient hypereosinophilia was observed 6 weeks after the start of daptomycin, in the absence of clinical and radiological signs of pneumonia, therefore we specify that we never diagnosed eosinophilic pneumonia. Despite that, we completely agree with the reviewer’s comment: we made our assumption on safety less strong, stating that the safety profile has not been determined yet, but we were able to treat the infection with clinical and microbiological cure before being forced to stop therapy due to a side effect of the medication. 

In the text, we modified as followed: “Daptomycin was otherwise effective and well-tolerated at 6 mg/kg twice a day and not to be delayed to ensure better cure rates. Its safety profile has not been determined yet, but we were able to treat the infection with clinical and microbiological cure, before being forced to stop therapy due to a side effect of the medication.”

 Otherwise i think this was well written and is useful for this population. 

We thank Reviewer 1 for this kind comment.

 The Echo pictures are a little disappointing. They may be the best you have, but better echo pictures would be better.

R1Q4: 

R1Q4: Unfortunately, these images were the best we could retrieve, directly downloaded from the ultrasound machine, We had already tried to enhance image quality by increasing to > 300 dpi in TIFF format.

Reviewer 2 Report

The authors presented a well written case report about the infective endocarditis in the extremely preterm newborns. In general, they could present in more details the some epidemiological data of this disease in the segment of introduction. Furthermore, there is a need for improvement of the morphology/presentation of the table of main studies. Also, they could present some details about the differential diagnosis in this patient population according to the week of life and the laboratory/clinical indices. Lastly, they could also present the clinical/laboratory data in a separate table. 

Author Response

Reviewer 2

The authors presented a well written case report about the infective endocarditis in the extremely preterm newborns.

We thank Reviewer 2 for this kind comment.

In general, they could present in more details the some epidemiological data of this disease in the segment of  introduction.

R2Q1: We added details about epidemiological data in the introduction, as suggested: “Its incidence is estimated to be 3.3 per 100.000 annually for infants younger than one year, with 0.05-0.12 estimated cases per 1000 pediatric hospital admissions”.

Furthermore, there is a need for improvement of the morphology/presentation of the table of main studies.

R2Q2: The morphology and presentation of the table of the main studies has been changed and improved, as suggested.

Also, they could present some details about the differential diagnosis in this patient population according to the week of life and the laboratory/clinical indices.

R2Q3: As suggested, we presented details about the differential diagnosis in this patient population (in the text “Possible differential diagnosis, considering the persistent positivity of blood cultures and elevated CRP levels, were sepsis by a different pathogen, or alternative foci of infection (abscess, meningitis, pneumonia, endocarditis”)

Lastly, they could also present the clinical/laboratory data in a separate table.

R2Q4: We added a new table (Table 2), presenting the patient’s clinical and laboratory data. We kept it simple in order not to be redundant, as some data are also partially presented in the timeline (including laboratory values).